# Updates on HPV Vaccination

**DOI:** 10.3390/diagnostics13020243

**Published:** 2023-01-09

**Authors:** Ojone Illah, Adeola Olaitan

**Affiliations:** Women’s Cancer Department, EGA Institute for Women’s Health, University College London, London WC1E 6BT, UK

**Keywords:** human papillomavirus, HPV, HPV vaccine, cervical cancer

## Abstract

Cervical cancer still poses a significant global challenge. Developed countries have mitigated this challenge by the introduction of structured screening programmes and, more recently, the HPV vaccine. Countries that have successfully introduced national HPV vaccination programmes are on course for cervical cancer elimination in a few decades. In developing countries that lack structured screening and HPV vaccination programmes, cervical cancer remains a major cause of morbidity and mortality. The HPV vaccine is key to addressing the disproportionate distribution of cervical cancer incidence, with much to be gained from increasing vaccine coverage and uptake globally. This review covers the history and science of the HPV vaccine, its efficacy, effectiveness and safety, and some of the considerations and challenges posed to the achievement of global HPV vaccination coverage and the consequent elimination of cervical cancer.

## 1. History of the HPV Vaccine

One of the most extraordinary stories in medical research is that of Henrietta Lacks, an African American woman who died from cervical cancer seventy years ago, at the age of thirty-one. “HeLa cells”, as they are commonly referred to, are the cells which were collected from her cervix in 1951 shortly before her death; these were the first established in vitro immortal cancer cell line [1] and have formed the basis for multiple advances in medical research, including the development of the human papillomavirus (HPV) vaccine.

In the originally biopsied HeLa cells, German virologist Harald zur Hausen in the early 1980’s discovered the presence of HPV-18 [2,3]. It is now known that HPV-18 and HPV-16 are responsible for 70% of cervical precancers and cancers. This ground-breaking discovery, for which zur Hausen won a Nobel prize in 2008, led to the fundamental research that is behind our detailed understanding of the natural history of HPV infection and its induced carcinogenesis [4]. It became established in 1999 that persistent HPV infection was a prerequisite for the development of most invasive cervical cancers [5].

HPV now forms the basis of both primary and secondary cervical cancer preventative measures. Most cervical screening programmes worldwide utilise HPV detection as a primary screening tool, in keeping with the World Health Organisation (WHO) recommendations [6]. HPV vaccine development was initiated in Australia in 1991; Dr Ian Fraser and Dr Jian Zhou developed viruslike particles (VLPs) based on proteins expressed on the HPV viral capsid [7]. These VLPs have proven essential for vaccine development.

The combined effect of HPV vaccination and cervical screening should effectively make cervical cancer a rare occurrence, yet it remains a leading cause of cancer deaths globally [8]. Worldwide, it is the fourth most common cancer that affects women while in low and low-and-middle-income countries (LLMICs), it is the second most common cancer affecting women. LLMICs account for 86% of the global cervical cancer burden, bearing the brunt of this challenge [8].

We review the natural history of HPV infection and its association with cervical cancer, the different types of HPV vaccine and studies looking into their efficacy, effectiveness and safety. We also review the implementation of the HPV vaccine globally, with a focus on the barriers faced by countries with a low vaccine uptake.

## 2. HPV and Cervical Carcinogenesis

HPV is a double-stranded DNA virus which belongs to the *Papillomaviridae* family. It is the most common sexually transmitted infection worldwide and it is estimated that most sexually active individuals will be infected with HPV at least once in their lifetime [9]. Infection is most prevalent in young individuals following the onset of sexual activity and in most countries, declines after the age of 35 [9] (Figure 1). Over 90% of exposed individuals will achieve immune-mediated spontaneous clearance of the virus within two years of exposure [10].

There are over 100 subtypes of HPV, characterised into high-risk and low-risk subtypes depending on oncogenic potential. Low-risk HPV subtypes include HPV-6, HPV-11, HPV-42, HPV-43 and HPV-44 [12]. HPV-6 and HPV-11 are the most prevalent nononcogenic subtypes and are responsible for over 90% of all cases of genital warts [13]. The oncogenic subtypes include HPV16, HPV-18, HPV-33, HPV-35, HPV-45 and HPV-58, and these have the potential to cause cervical, oropharyngeal, vaginal, vulvar, penile and anal cancers [12,14]. Cervical cancer is by far the most prevalent HPV-associated cancer, and HPV 16 is the most common causative subtype, followed by HPV-18 [14]. Jointly HPV-16 and HPV-18 account for approximately 70% of all cervical cancer cases [9].

There may be racial and ethnic differences in age at peak HPV incidence, degree of HPV persistence and prevalence of high-risk HPV subtypes [14,15,16] and a number of studies have looked into these differences by region. Some recent studies have suggested a higher burden of non-HPV-16 or HPV-18 in Black and African women with cervical cancer [17,18]. While this requires further studies, HPV-16 and HPV-18 remain the most common high-risk subtypes associated with cervical cancer worldwide [14,19].

### HPV-Induced Cervical Carcinogenesis

The HPV genome consists of three regions. The long control region (LCR) regulates gene expression and replication, the early (E) region encodes for proteins which are required for HPV gene expression, replication and survival, and the late (L) region encodes for viral structural proteins [14]. The virus gains access to host cells by infecting the basal epithelial layer of the cervix. Following infection, early proteins E1, E2, E4, E5, E6 and E7 are produced; E6 and E7 are the key oncoproteins which promote viral DNA replication and prevent apoptosis via interactions with tumour suppression proteins [9,14]. Subsequently, late capsid proteins L1 and L2 are produced, which allow the formation of progeny virions in host nuclei cells—these replicate the viral life cycle [9] (Figure 2).

The majority of women infected with HPV will clear it within a few years. An estimated 10–20% of women will have persistent HPV infection [20], which if untreated, can lead to cervical cancer after 15 to 20 years. In immunosuppressed states such as with untreated HIV, cancer can develop in 5 to 10 years [10]. The persistence of HPV infection occurs via (i) malignant transformation by the HPV oncoprotein-mediated downregulation of cell cycle control and genetic damage, and (ii) a wide array of immune evasion mechanisms which high-risk papillomaviruses have developed [9,21]. From persistent infection, cervical dysplasia and cervical intraepithelial neoplasia (CIN) can develop, which may progress to invasive cervical cancer (Figure 2).

## 3. Prophylactic HPV Vaccine Types and Mechanism of Action

This understanding of HPV-induced carcinogenesis alongside the availability of VLPs has allowed various prophylactic HPV vaccines to be developed. The first to be licensed in 2006 was Gardasil^®^ (Merck, Sharp & Dome (Merck & Co., Whitehouse Station, NJ, USA)), a quadrivalent vaccine which targets HPV-6, HPV-11, HPV-16 and HPV-18, conferring protection against genital warts, which are most commonly caused by HPV-6 and HPV-11. In 2007, a bivalent vaccine, Cervarix^®^ (GlaxoSmithKline, Rixensart, Belgium), was licensed, which targets HPV-16 and HPV-18. Gardasil 9^®^ (Merck, Sharp & Dome (Merck & Co., Whitehouse Station, NJ, USA)) is a nonavalent vaccine licensed in 2014 that targets HPV-6, HPV-11, HPV-16, HPV-18, HPV-31, HPV-33, HPV-45, HPV-52 and HPV-58 [21,22]. These three established vaccines utilise eukaryotic cells as producer cells. More recently, a bivalent vaccine, Cecolin^®^ (Xiamen Innovax Biotechnology, Xiamen, China), was developed, which utilises *Escherichia coli* to produce HPV-16 and HPV-18 L1 VLPs [23]. Cecolin^®^ was licensed for use in China in 2020 and prequalified by the WHO in 2021. Finally, a recombinant bivalent HPV vaccine (Shanghai Zerun Biotechnology, subsidiary of Walvax Biotechnology, Shanghai, China) targeting HPV-16 and HPV-18 was also licensed for use in China in 2022 and prequalified by WHO in 2022 [24]. Table 1 summarises the characteristics of current WHO-prequalified vaccines, however the HPV vaccine development market remains active, and a quadrivalent vaccine, Cervavac, was recently launched in India [25].

The vaccines have a similar mode of action based on VLPs, a recombinant, noninfectious assembly of the L1 HPV capsid protein. VLPs are antigenically identical to infection-causing HPV virions [26], such that exposure to VLPs induces a strong neutralising antibody response, which halts HPV uptake by the basal epithelial cells of the cervix. This humoral response is responsible for the efficacy of the vaccine. The vaccine mode of action also explains why the HPV vaccine is less effective in women with prior HPV exposure, as HPV infection of the basal cells has already occurred [27,28]. Each vaccine has subtype-specific VLPs, depending on their targets; however, they all confer a degree of cross-protection against other, nonvaccine HPV subtypes [29]. It is on this basis that the WHO considers all vaccines to be equally protective against cervical cancer [10].

## 4. Prophylactic HPV Vaccine Clinical Trials

Prior to licensure, all HPV vaccines underwent clinical trials to determine efficacy, effectiveness and safety. A WHO convention in 2003 determined appropriate HPV vaccine trial endpoints: it was agreed that ethical and time constraints could not allow for cervical cancer to be an appropriate trial endpoint, given that participants under follow-up during the trial period would receive treatment of any cervical precancerous lesions detected [30]. Agreed study endpoints included Grade 2 and above CIN (CIN2+), and HPV infection [30], as these are sufficient to demonstrate vaccine efficacy at reducing persistent HPV infection and cervical precancer. After licensure, multiple HPV vaccine trials were performed, summarised in a systematic review published in 2018. This review included 26 randomised control trials and concluded that the vaccine was highly efficacious at preventing cervical precancer [31]. The key prelicensure trials are discussed below.

### 4.1. Clinical Trials in Younger Females

The phase 3 efficacy trials conducted for HPV vaccines all included younger women aged 15 to 26. These trials were multinational and included thousands of participants, with the trial design and conduct led by the respective manufacturers. The results from these trials are summarised in Table 2.

#### 4.1.1. The Quadrivalent HPV Vaccine—Gardasil^®^

The FUTURE I and II trials assessed the efficacy of the Gardasil^®^ quadrivalent HPV vaccine, with results published in 2007 [32]. Over 12,000 female participants aged 15 to 26 were randomised to receive either three doses of the quadrivalent vaccine or a placebo. After a three-year follow-up period, vaccine efficacy against HPV-16- and HPV-18-associated high-grade cervical disease was 98% in women without prior exposure to HPV-16 or HPV-18. Vaccine efficacy in all participants, including those with prior HPV-16 or HPV-18 exposure, was 44% [32].

A subset of European participants from the FUTURE II trial were followed up for fourteen years, with no cases of HPV-16- or HPV-18-associated high-grade cervical dysplasia noted in follow-up period (vaccine efficacy of 100%) [33]. Seropositivity for all four HPV subtypes at the fourteen-year follow-up mark remained high at >90% [33].

A double-blinded randomised trial of the quadrivalent HPV vaccine in a Japanese population of women aged 18 to 26 similarly found a high efficacy against vaccine-type high-grade cervical disease [34].

#### 4.1.2. The Bivalent HPV Vaccine—Cervarix^®^

The PATRICIA trial was the first to assess the efficacy of the Cervarix^®^ bivalent HPV vaccine [35,36]. Over 18,000 women aged 15 to 25 were recruited and randomised to receive either three doses of the Cervarix^®^ vaccine or the hepatitis A vaccine. After a three-year follow-up period, vaccine efficacy against HPV-16 or HPV-18-associated high-grade CIN was 92.9% [36]. A subgroup analysis showed that the efficacy against vaccine-type CIN3+ was as high as 100% in the HPV-naïve cohort [37].

The Costa Rican Vaccine Trial (CVT) was a second, large prelicensure trial investigating the efficacy and safety of Cervarix^®^. After a four-year follow-up period, vaccine efficacy was 89% against HPV-16- or HPV-18-associated CIN2+ [38]. A longer-term follow-up at 11 years showed a vaccine efficacy of 100% against HPV-16- or HPV-18- associated CIN2+ [39].

The Cervarix^®^ vaccine was also studied in a population of >6000 Chinese women aged 18 to 25 showing a vaccine efficacy of 100% in HPV-naïve participants [40].

#### 4.1.3. The Nonavalent HPV Vaccine—Gardasil 9^®^

The efficacy of the nonavalent Gardasil^®^ 9 vaccine was assessed in a multinational double-blind trial in women aged 16 to 25 [41]. Over 14,000 participants were recruited, and unlike the other trials, the control group received the quadrivalent Gardasil^®^ HPV vaccine. Vaccine efficacy against high-grade cervical disease associated with HPV-31, HPV-33, HPV-45, HPV-52 and HPV-58 (i.e., HPV subtypes not covered by the quadrivalent vaccine) was 97.1%; the number of cases per 10,000 person-years was 0.5 in the nonavalent group versus 18.1 in the quadrivalent group. The incidence of abnormalities associated with HPV-6, HPV-11, HPV-16 and HPV-18 (i.e., HPV subtypes covered by the quadrivalent vaccine) was comparable between both groups of participants. Furthermore, the immunogenicity of the nonavalent vaccine with respect to HPV-6, HPV-11, HPV-16 and HPV-18 was comparable to that of the quadrivalent vaccine [41], with antibody response persisting for up to five years [42]. The authors concluded that the nonavalent vaccine could potentially prevent more cervical cancer cases by providing a broader coverage with a sustained high efficacy against all vaccine HPV subtypes.

#### 4.1.4. The Bivalent Cecolin^®^ Vaccine

The efficacy trial for the bivalent Cecolin^®^ vaccine recruited over 7000 female participants aged 18 to 45, in multiple centres in China between 2012 and 2013. The control group received the hepatitis E vaccine. Vaccine efficacy against high-grade genital disease and persistent infection associated with HPV-16 or HPV-18 were 100% and 97.3%, respectively [43,44]. Participants were age-stratified into two groups: 18–26 and 27–45, allowing a subgroup analysis. A phase 3 clinical trial of the Cecolin^®^ vaccine is ongoing in Bangladesh and Ghana, with results expected in 2023 (NCT04508309).

**Table 2 diagnostics-13-00243-t002:** Summary of prelicensure trials on HPV vaccine efficacy in younger females.

Trial	Number of Participants	Participant Ages	Efficacy against Vaccine-Type CIN2+
**Gardasil^®^**			
FUTURE I and II [33] NCT00092521 and NCT00092534	12,167	15–26	98%
**Cervarix^®^**			
PATRICIA [37] NCT00122681	18,644	15–25	92.9%
CVT [39] NCT00128661	7466	15–25	89.5%
**Gardasil 9^®^**			
NCT00543543 [42]	14,215	16–25	97.1%
**Cecolin** ** ^®^ **			
NCT01735006 [45]	3723	18–26	100%

### 4.2. Clinical Trials in Older Females

The quadrivalent Gardasil^®^ vaccine and the bivalent Cervarix^®^ vaccine have both been trialled in older females, a group likely to have reduced efficacy since vaccine efficacy declines with prior HPV exposure. The FUTURE III trial evaluated the efficacy of the quadrivalent vaccine in over 3800 women aged 24 to 45. Vaccine efficacy against vaccine-type CIN1+ was 88.7% in HPV-naïve women and 30.9% in all women [45,46]. Another double-blinded trial of the quadrivalent vaccine in Chinese women aged 20 to 45 found a high vaccine efficacy of 94% [47]. The VIVIANE trial evaluated the bivalent Cervarix^®^ vaccine’s efficacy in women aged > 25. Vaccine efficacy against combined endpoint of vaccine-type 6-month persistent infection and CIN1+ was 90.5% in the per protocol group and 86.5% in the total vaccinated cohort. Estimated vaccine efficacy against vaccine-type CIN2+ was high but insignificant due to low numbers [48].

While these studies support the use of the HPV vaccine in older females, the effectiveness of the vaccine in this group is less than in adolescents: because the incidence of HPV infection declines with age (Figure 1), older females are less likely than adolescent females to develop new HPV infection which reduces the cost-effectiveness of routine vaccination in this group. Thus, in the UK, HPV vaccination of women >25 can only be obtained privately [49], whereas the US recommends shared decision-making with clinicians for women aged between 27 and 45 [50].

### 4.3. HPV Vaccine with Previous Known Infection

In many countries, there has been a move to HPV-based cervical screening, meaning that many women will already be known to have had a previous HPV infection. It is therefore important to define the efficacy of the HPV vaccine in this group. Most of the HPV vaccine clinical trials included in the total vaccinated cohort participants who, at the time of recruitment, were positive for HPV DNA or seropositive for HPV antibodies. Unsurprisingly, these groups had lower vaccine efficacy compared to HPV-naïve participants. Vaccine efficacy in seropositive, DNA-negative participants, i.e., those with previous but no current HPV infection, was summarised in a meta-analysis which pooled data from trials on the bivalent Cervarix^®^ vaccine and the quadrivalent Gardasil^®^ vaccine [51]. Against CIN2+, the pooled HPV vaccine efficacy was 85%, and against persistent serotype-specific HPV infection, the pooled vaccine efficacy was 78% (six months of persistent infection) and 80% (twelve months of persistent infection) [51]. These findings support the use of the vaccine in HPV-DNA-negative women, regardless of their serostatus.

### 4.4. HPV Vaccine in HIV Infection

People with HIV are known to have lower levels of immunity, which make them more susceptible to persistent HPV infection. A few studies have investigated the efficacy and safety of the HPV vaccine in this population, showing a 100% seroconversion rate following vaccination with no adverse outcomes [52]. However, dedicated HPV vaccine efficacy data are lacking in this population and require further assessment.

### 4.5. HPV Vaccination in Males

In males, HPV is associated with a range of anogenital and oropharyngeal diseases—over 90% of anal cancers, over 70% of oropharyngeal cancers and up to 48% of penile cancers are associated with HPV [53]. HPV vaccination of males therefore has the potential to reduce HPV infection and associated lesions. A randomised controlled trial of over 4000 males aged 16 to 26 found a quadrivalent vaccine efficacy of 90.4% against vaccine-type anogenital lesions and 85.6% against persistent vaccine-type HPV infections. Similar to women, vaccine efficacy was reduced in participants with HPV infection at baseline [54]. The long-term follow-up study from this trial was able to assess primary outcomes of genital warts, vaccine-type external genital lesions and vaccine-type anal intraepithelial neoplasia or anal cancer in men who have sex with men (MSM) [55]. Vaccine efficacy against vaccine-type genital warts and external genital lesions was 89.9% and 90.8%, respectively, in the HPV naïve group. In the intention-to-treat group, vaccine efficacy against vaccine-type external genital lesions was 66.7% [56]. In MSM, vaccine efficacy against vaccine-type anal intraepithelial neoplasia and anal cancer was 89.6% in the HPV naïve group and 50.3% in the intention-to-treat group [56]. Studies have not yet reported on vaccine efficacy against head and neck cancers and penile cancers.

Unlike the quadrivalent Gardasil^®^ vaccine, the nonavalent Gardasil 9^®^ vaccine did not undergo clinical trials in men prior to licensure. Rather, immunogenicity studies were done which showed that the nonavalent vaccine elicited immune responses similar to the quadrivalent vaccine against HPV-6, HPV-11, HPV-16 and HPV-18 [57]. Based on this and other safety data, Gardasil 9^®^ has been licensed for use in men [58]. In the US, the FDA approved its indicated use in the prevention of HPV-associated oropharyngeal and head and neck cancers; a randomised controlled trial is underway to evaluate the efficacy of the Gardasil 9^®^ vaccine against persistent oral HPV infection [59].

### 4.6. Cross-Protection against Nonvaccine HPV Subtypes

The prelicensure HPV vaccine trials explored protection conferred by the vaccines against nonvaccine subtypes, i.e., cross-protection, with all vaccines demonstrating partial cross-protection against nonvaccine-type infection and disease. The FUTURE I and II trials of the quadrivalent vaccine assessed cross-protection against ten other HPV subtypes—31, 33, 35, 39, 45, 51, 52, 56, 58 and 59. These showed some cross-protective effect against HPV infection and low-grade CIN, most marked against HPV-31, but not against high-grade CIN [29].

The PATRICIA and Costa Rica trials of Cervarix^®^ also assessed the cross-protective effects of the bivalent vaccine against nonvaccine-type persistent infection and CIN2+ [60]. A high vaccine efficacy was noted against infection and CIN2+ associated with HPV-31, HPV-33 (both subtypes closely related to HPV-16), HPV-45 (closely related to HPV-18) and HPV-51. Vaccine efficacy was higher in women who were HPV-naïve at baseline [60,61]. Long-term follow-up data from Cervarix^®^ trials indicate cross-protective effect lasting at least 11 years [62,63].

## 5. HPV Vaccine Safety

As with all vaccines, the safety of the HPV vaccine was evaluated in prelicensure trials and continues to be evaluated in surveillance systems worldwide following licensure. Data on HPV vaccine safety are robust and have consistently shown no concerns [64,65,66,67]. Injection site reactions such as pain, swelling and redness were the most commonly reported adverse events following vaccine administration, with a slight increased frequency of these reactions reported with the use of the bivalent Cervarix^®^ and nonavalent Gardasil^®^ vaccines [64]. In vaccine clinical trials, the rates of systemic adverse events were similar between vaccine and control groups; fever, nausea, headache and dizziness were the most commonly reported systemic adverse events [64]. Following the recognition that syncope could occur after vaccination, recommendations were made in the US for adolescents to be seated during vaccination [68].

In vaccine clinical trials, there was no difference in rates of serious adverse events between vaccine and control groups [64,67]. National surveillance data following the introduction of the bivalent Cervarix^®^ and quadrivalent Gardasil^®^ vaccines in various countries showed a low rate of serious adverse events [64]. Most of the available surveillance data from the nonavalent vaccine are from the US, which have shown that the nonavalent vaccine has a similar safety profile to the quadrivalent vaccine [68]. There have been no reported deaths attributable to the HPV vaccine [64,68].

The HPV vaccine is not currently recommended in pregnancy, but safety data from this group have been obtained from prelicensure trials, clinical trials and surveillance systems where women were vaccinated during pregnancy or became pregnant shortly after vaccination. None of these found any association between vaccine administration and adverse pregnancy outcomes including spontaneous abortion, foetal loss and congenital abnormalities [64]. Furthermore, the prelicensure trials found similar rates of pregnancy between vaccinated and unvaccinated groups, and though there have been several case reports linking the vaccine to primary ovarian insufficiency, the surveillance data to date have been unable to establish this as a causal association [64].

Finally, the safety of the vaccine with regards to the development of autoimmune disease, venous thromboembolism and neurological conditions has been studied in various contexts, with no association found between the HPV vaccine and these conditions [64]. The evidence all points towards the safety of the vaccine, as has been supported by the WHO [69], the Global Safety Vaccine Advisory Group [70] and various national and international immunisation advisory committees.

Despite these robust safety data, concerns about vaccine safety have impacted vaccine uptake in several countries. A notable example of this is Japan, who introduced HPV vaccination into their routine national immunisation schedule in April 2013. Due to physical symptoms including pain and motor impairment reported by vaccinated girls, the vaccination programme was suspended ten weeks later in June 2013. The vaccine remained available; however, its proactive recommendation was discontinued causing a reduction in vaccine coverage from 70% to less than 1% [71]. A modelling study showed the effect of this is an additional (projected) 25,000 preventable cervical cancer cases and 5000 cervical cancer deaths in Japan [71]. Real-world data have shown the effect of this in 2020 has been an increase in HPV-16/HPV-18 infection rates compared to the previous years that featured vaccinated cohorts [72]. After a nine-year hiatus, the Japanese health ministry has reversed its position on this and, as of April 2022, has restarted the active recommendation of the vaccine to 12-to-16-year-old girls [73]. It is hoped that with increasing safety data available, public confidence in the safety of the HPV vaccine will be strengthened.

## 6. Impact of HPV Vaccination on Populations

In the initial ten years following the HPV vaccine introduction, several population-level studies assessed the real-world effectiveness of the vaccine on HPV-16/HPV-18 infection, anogenital warts and high-grade CIN. In 2019, findings from these were summarised in a systematic review and meta-analysis [74]. Data from 65 studies in 14 countries were included showing significant age-dependent reductions in HPV-16 and HPV-18 prevalence (83% reduction in 13-to-19-year-old girls, and 66% reduction in 20-to-24-year-old women), anogenital warts in both males and females and CIN2+ (51% reduction in 15-to-19-year-old girls and 31% reduction in 20-to-24-year-old women) [74].

Ultimately, the aim of the HPV vaccine is to reduce the burden of HPV-associated cancers, and with longer-term population data increasingly emerging, a few studies have been able to report on this outcome. Sweden introduced the quadrivalent Gardasil^®^ HPV vaccine nationally in 2009, and was one of the first countries, in 2020, to report on cervical cancer outcomes following vaccine introduction. Their national data showed a remarkable 88% reduction in cervical cancer incidence in women who had been vaccinated prior to age 17, and a 53% reduction in cervical cancer incidence in women vaccinated between the age of 17 and 30 [75].

In the UK, the bivalent vaccine was introduced in England in 2008, and data on vaccine effectiveness against cervical cancer were published in 2021. Population data from England showed an 87% reduction in cervical cancer incidence in women vaccinated at age 12 to 13, a 62% reduction in women vaccinated at age 14 to 16 and a 34% reduction in women vaccinated at age 16 to 18 [76]. The authors estimated that there were 448 fewer than expected cervical cancers and over 17,000 fewer than expected CIN3 among the vaccinated cohort in England (Figure 3) [76].

Another national study in Denmark published in 2021, where the HPV vaccine was introduced into the childhood vaccination programme in 2009, reported a high vaccine effectiveness against cervical cancer. The incidence rate ratio was 0.14 in females vaccinated under the age of 17, and 0.32 in females vaccinated between age 17 and 20, when compared to unvaccinated females. Mirroring vaccine efficacy data, the effectiveness of the vaccine was reduced when administered at an older age [77].

## 7. Number of Doses

HPV vaccines were initially licensed with a recommended three-dose schedule, and the prelicensure trial protocols included three vaccine doses, which demonstrated a high efficacy against vaccine-type CIN2+ and vaccine-type persistent HPV infection. This led to a three-dose regimen being recommended by the WHO and national governing bodies.

The Costa Rica Vaccine trial group, who conducted a prelicensure trial for the Cervarix^®^ bivalent vaccine, carried out a post hoc analysis investigating the vaccine efficacy in women who received less than three doses of the bivalent HPV vaccine. Interestingly, they found that women who received less than three doses of the vaccine had a similar vaccine efficacy against persistent HPV-16/HPV-18 infection, relative to the women who received the per-protocol three doses; this protection lasted for up to ten years [78,79,80]. Although vaccine efficacy was similar, a lower, albeit stable, antibody response was noticed in the group that received a single dose of the vaccine [81].

These findings were of huge significance, as a reduced dosing schedule has important public health implications, easing both financial and logistical barriers to the introduction of HPV vaccination programmes. A single-dosing schedule would significantly reduce supply and storage issues and increase compliance. Even with a modest assumed single-dose vaccine efficacy of 80%, there would be (projected) millions of averted cases of cervical cancer [82]. Thus, several trials have followed on from the Costa Rica Vaccine trial to formally investigate HPV vaccine efficacy using two-dose and one-dose schedules.

A group In India, the Indian HPV vaccine study group, designed a randomised trial to compare vaccine efficacy between one-, two- and three-dosing schedules. Trial plans were interrupted in 2010 when the Indian government suspended HPV vaccination. Consequently, the design was changed to an observational cohort study consisting of >12,000 participants aged 10 to 18. There were over 4000 participants in each dose category [83]. Early data showed that although a single dose resulted in lower antibody titres compared to two or three doses, immune response was sustained and stable over a four-year period [84]. Ten-year follow-up data from the group showed that the vaccine efficacy against HPV-16/HPV-18 infection was similar in participants receiving one, two or three doses at 95%, 93% and 93%, respectively (Figure 4) [85].

Another study, the ESCUDDO trial, is an ongoing randomised double-blind design comparing the vaccine efficacy of a one- versus two-dosing schedule of the bivalent and the nonavalent HPV vaccines [86]. Over 20,000 participants were recruited aged 12 to 16 with the primary objectives of comparing HPV-16/HPV-18 infection rates between the one-dose and two-dose groups and estimating vaccine efficacy for a one-dose regimen [86]. For ethical reasons, a placebo arm was not included, and instead, a survey of unvaccinated participants was used to estimate vaccine efficacy. Results from this trial are expected in 2024.

Similar randomised controlled trials are ongoing in various African countries to investigate the efficacy of a single-dosing vaccine schedule including the DoRIS trial in Tanzania (NCT02834637), the HANDS trial in Gambia (NCT03832049) and the KEN SHE trial in Kenya (NCT03675256) [87]. Preliminary results from the KEN SHE trial showed the high vaccine efficacy of a single dose of the bivalent (97.5%) and nonavalent vaccines (97.5%) at preventing persistent HPV-16/18 infection after eighteen months [88]. Observational studies in various countries have additionally shown the high efficacy of single-dose HPV vaccination schedules [89,90,91]. Furthermore, data from the Costa Rica vaccine trial group and the Indian HPV vaccine study group will continue to provide long-term outcomes.

With the evidence pointing towards the high efficacy of a single dose of the HPV vaccine, the Joint Committee on Vaccination and Immunisation (JCVI) in the UK has recently recommended a single HPV vaccine dose in their routine adolescent and MSM vaccination programmes [92]. The WHO Strategic Advisory Group of Experts on Immunisation (SAGE) also met in April 2022 to review the evidence on single-HPV-vaccine dose efficacy. The outcome from this was a recommendation for single-dosing schedules for some low-risk groups; it is likely the WHO will update their recommendation following a review of this [93]. In the Unites States, the Advisory Committee on Immunization Practices (ACIP) are yet to change from the current recommended two-dose schedule [50].

## 8. Prophylactic HPV Vaccine as Adjunct Treatment for CIN

Even following treatment for CIN, women remain susceptible to reinfection with HPV and have been shown to be at increased risk of developing recurrent CIN and HPV-associated cancers [94,95]. There is growing evidence from a few studies that the use of prophylactic HPV vaccines in women having a treatment for CIN may reduce the risk of recurrent disease. The SPERANZA study was the first prospective study to assess the effectiveness of the HPV vaccine for women undergoing surgical management of high-grade cervical disease. Results showed an 81% reduced risk of HPV-associated high-grade recurrent disease in vaccinated women [96]. Several other studies followed, including the VENUS study, another observational study which compared outcomes between vaccinated and unvaccinated women who underwent conization for CIN2-3. Results showed a 59% reduction in persistent/recurrent CIN2-3 in vaccinated women [97]. Two randomised control trials also showed reduced CIN recurrence rates in vaccinated women [98,99].

A systematic review was recently published, which evaluated 22 studies reporting on HPV infection and the risk of HPV-associated disease following surgical management of HPV-associated genital disease in vaccinated individuals [100]. Although authors concluded that there might be a reduction in the risk of recurrent high-grade cervical disease with prophylactic HPV vaccination use, they highlighted the requirement for further adequately powered randomised control trials, to establish vaccine use in this setting [100]. The NOVEL trial (NCT03979014) is an ongoing randomised control trial comparing outcomes between a group of women receiving local treatment plus vaccination with the nonavalent HPV vaccine, and a group receiving local treatment only. It is anticipated that results from this trial will give insight into the effectiveness of the HPV vaccine as an adjunctive treatment in women undergoing surgical excision of HPV-associated premalignant cervical disease.

## 9. HPV Vaccination Implementation, Programmes and Coverage

Since HPV vaccines became available in 2006, countries have progressively introduced the vaccine into their national immunisation schedules. The WHO initially recommended HPV vaccines in 2009, using a three-dosing schedule to girls aged 9–14 years [101]. In 2014, this recommendation changed to a two-dosing schedule [102] and further evolved in 2017 to recommend vaccination of multiage cohorts [69], although this was temporarily paused in 2019 following vaccine supply issues.

As of March 2022, 60% of WHO member states have introduced the HPV vaccine into their national immunisation schedule (Figure 5). The majority of these are high-income and upper-middle-income countries [24], and some of the most populous nations are yet to introduce HPV vaccination into their immunisation schedules. The consequence of this is that global coverage of the HPV vaccine remains low at 12% for two doses in females, as of 2020 [103].

Strategies to deliver the HPV vaccine include school-based programmes, healthcare facility-based programmes, and outreach/campaign programmes. School-based HPV vaccination programmes have been successful at achieving a high HPV vaccine coverage as demonstrated in several countries. Australia was one of the first countries to implement a national HPV vaccination programme in 2007. This was a government-funded, school-based immunisation programme, offering three doses of the quadrivalent vaccine to 12-to-13-year-old girls. In 2013, the programme was expanded to include boys, and in 2018, changed to two doses of the nonavalent vaccine [104]. Since the vaccine introduction, Australia has maintained a high vaccination coverage, and if this is maintained, is projected to eliminate cervical cancer (age standardised incidence of <4 new cases per 100,000 women) by 2028 [105].

The UK similarly introduced a government-funded school programme aimed at 12-to-13-year-old girls in 2008. The triple dosing schedule was changed to a two-dose schedule in 2014, and the vaccination of 12-to-13-year-old boys was included in 2019 [92]. Prior to the COVID-19 pandemic, the HPV vaccine coverage was high at >80%; the COVID-19 pandemic disrupted school-based vaccinations as school attendance dropped, and although improving, the vaccine coverage is not yet back to prepandemic levels [106].

In the United States, routine HPV vaccination for girls aged 11 to 12 was recommended as a three-dosing schedule in 2006. This also included catch-up vaccination for women up to the age of 26. In 2011, routine vaccination was recommended for males aged 11 to 12. In 2016, the standard dosing schedule was reduced to two doses [107]. HPV vaccination in the United States is delivered mainly in primary care and healthcare facilities; Although a coverage of 75% has been attained [108], there are significant variations in vaccine coverage dependent on race, ethnicity and socioeconomic status within the United States [109].

Unsurprisingly, many LLMICs have not been able to introduce the HPV vaccine into national immunisation schedules, due to financial and infrastructural constraints. As of March 2022, 114 out of 145 (78.6%) high-income and upper-middle-income countries have introduced HPV vaccination, whereas only 20 out of 80 (37.5%) low-income and low-middle-income countries have introduced HPV vaccinations nationally [110]. Many of these countries consequently have a low HPV vaccination coverage. In these countries where engagement with healthcare services is often minimal, school-based vaccination programmes would likely achieve the highest levels of coverage. Unfortunately, this approach is unsustainable due to funding issues, and the majority of HPV vaccine delivery has been via campaign approaches [111]. A notable exception to this is Rwanda, an LLMIC which has been able to introduce a successful national HPV vaccination programme. Rwanda was the first African country to introduce a national HPV vaccination programme in 2011, which was via a school-based approach financially supported by Merck. They have consequently been able to attain a high vaccine coverage of over 90% [112,113].

In Asia, several countries have been able to add HPV vaccination to their national programmes, such as Thailand, Sri Lanka, Bhutan and the Maldives [25]. In China and India, the HPV vaccine is licensed and available, and there are plans to introduce the vaccine to routine national immunisation programmes from 2024 onwards [24].

As of 2022, gender-neutral HPV vaccination schedules are present in 39 WHO member states, and many high-income countries have shifted/are shifting into an era of equity of access to HPV vaccination for all genders [110,114]. Due to the HPV vaccine shortage in recent years, the WHO has not yet endorsed the vaccination of males and have instead recommended that the vaccination of boys be suspended temporarily to relieve supply constraints [115]. Opinions on this are mixed, where on the one hand, the benefit of male vaccination is evident in the reduction of HPV infection and its associated sequelae. On the other hand, however, the health benefit of vaccinating more girls in LLMICs, where rates of cervical cancer are much higher and screening measures less well established, is just as, if not more, impactful.

## 10. Barriers to HPV Vaccine Uptake

The impact of the HPV vaccine on cervical cancer incidence can only be realised with a high vaccine coverage, and several barriers continue to hinder this. Unfortunately, the countries with the highest cervical cancer burden and therefore in most need of the HPV vaccine face the greatest barriers. Addressing these barriers are key to achieving the full benefit of the vaccine.

### 10.1. Cost

Vaccine cost is a significant barrier, especially in LLMICs where the introduction of national HPV vaccination programmes is not financially feasible without external support. Gavi, the vaccine alliance, is a global public–private partnership which works with manufacturers to reduce the cost of vaccines to burdened countries, particularly LLMICs. They introduced a programme in 2012 to enable Gavi-eligible countries to introduce HPV vaccination programmes nationally at a negotiated cost of <$5 per dose, versus the usual >USD 100 per dose [116]. There were two ways of accessing Gavi’s support for national vaccine introduction: countries either needed to have experience in the delivery of a multidose adolescent vaccination programme, or they first had to demonstrate their readiness to deliver national HPV vaccination via a two-year demonstration project [116,117]. This unfortunately left very few countries eligible for the introduction of a national HPV vaccination programme, until at least 2015, when the two-year demonstration project would have been complete. In this period, the national vaccination introductions that occurred in LLMICs were funded by donations from pharmaceutical companies or nongovernmental organisation (NGO) support [116]. Gavi changed its policy in 2016, allowing eligible countries to apply directly for a national HPV vaccination introduction, but shortly after this, a worldwide vaccine supply issue ensued [118].

### 10.2. HPV Vaccine Shortage

Since 2018, there has been a worldwide shortage of HPV vaccines resulting in significant supply constraints. This resulted in adjustments to planned vaccine introduction programmes, which particularly affected LLMICs. The shortages also led WHO’s SAGE to suspend recommendations to vaccinate males and multiage cohorts, in order to reduce the impact of the vaccine shortage [24]. These adjustment and suspensions will likely affect LLMICs for years to come. As previously highlighted, this has also brought into moral question the addition of males to HPV vaccination programmes, which has been done successfully in several high-income countries. In 2022, the WHO has prequalified two additional bivalent vaccines, Innovax’s Cecolin^®^ vaccine and Walvax Biotechnology’s vaccine, and it is anticipated that this, alongside a ramp-up of production of currently licenced vaccines, will help reverse vaccine shortages by 2023 [24,118].

### 10.3. Cold-Chain Requirement

Another huge barrier to HPV vaccination which considerably affects LLMICs is the requirement for cold-chain preservation for current HPV vaccines, which adds to already high costs. One way to address this is via lyophilization, which would dehydrate vaccine components and allow transfer of a frozen powder form at higher temperatures [119]. Alternatively, heat-stable capsomer preparations would permit more fluctuation in temperature and therefore greatly reduce costs [53]. Unfortunately, neither one of these preparations are available at present.

### 10.4. Adolescent Age Group

The HPV vaccine is targeted at adolescent/school-age children, who are a harder group to reach compared to the age group targeted by childhood immunisation programmes. The use of school-based vaccination programmes helps to circumvent this; however, many LLMICs do not have funded school health programmes, which require additional costs and manpower [111]. Furthermore, with school dropout rates higher in LLMICs [111], many adolescents could get missed from school-based programmes. Adding HPV vaccination to childhood immunisation programmes and/or coadministration with other vaccines are proposed solutions to this barrier [120]. Several studies have already shown that the coadministration of the HPV vaccine with other vaccines is safe and immunogenic [121] and a study is to begin shortly, investigating the immunogenicity of Gardasil 9^®^ in children aged 4 to 8 years (NCT05329961).

### 10.5. COVID-19 Pandemic

Globally, the COVID-19 pandemic has affected existing HPV vaccination programmes and halted the introduction of new programmes via the closure of schools and suspension of routine immunisation programmes. This has affected other vaccination programmes, and many are still recovering from the impact of the pandemic [120].

### 10.6. Vaccine Hesitancy

Vaccine hesitancy has a huge impact on HPV vaccine uptake globally. Misinformation on safety concerns about the vaccine, and an unfounded association of HPV vaccine use with sexual promiscuity are common drivers behind vaccine hesitancy [120]. A combination of culturally sensitive education and government-driven health policies can help improve public confidence and vaccine acceptance.

## 11. Therapeutic HPV Vaccines

Despite the high efficacy of prophylactic HPV vaccines, worldwide prevalence of high-risk HPV infection remains high, mainly due to the low global vaccine uptake and pre-existing HPV infection. This means the elimination of HPV infections and associated diseases could take decades still. Due to their mechanism of action, by inducing a humoral response to prevent an initial HPV infection, prophylactic HPV vaccines cannot have a reliable therapeutic effect [122]. This has led to a huge research effort towards the development of therapeutic HPV vaccines. These vaccines work by stimulating cell-mediated immunity, versus humoral-mediated immunity, against existing HPV infections and lesions. Many of these vaccines target the E6 and E7 oncoproteins, which are continuously expressed in premalignant and invasive lesions and are essential for cell-cycle arrest and the onset and progression of malignancy [123]. Vector types for therapeutic vaccine development have included DNA/RNA-based, peptide-based, protein-based and bacterial and viral vectors [122].

There are no licensed therapeutic vaccines yet, although several have undergone phase 2 and 3 clinical trials for use in the treatment of CIN and cervical cancer. VGX-3100 is one of such, a DNA-based therapeutic vaccine targeting the E6 and E7 proteins of HPV-16 and -18. Results from a phase 2 clinical trial showed histopathological regression from CIN2/3 to CIN 1 in 49% of vaccinated participants at six months [124]. None of the participants who regressed at six months had high-grade cytology findings eighteen months following vaccination, and 91% had no detectable HPV-16/18 infection [125]. The VGX-3100 vaccine is currently in phase 3 clinical trials (NCT03185013 and NCT03721978). Bacterial vector therapeutic vaccines such as the lactobacilli-based IGMKK16E7 [126] and the *Listeria*-based ADXS11-001 [127] are in late clinical trial phases, having shown similar promising results in earlier phase trials. Of particular note is the ADXS11-001 vaccine, which showed a 12-month survival of 38% when used in the treatment of advanced cervical cancer; this is significantly higher than historical survival of 25% [127].

In trial settings, therapeutic HPV vaccines have also been combined with other anticancer agents, such as antibodies against programmed death-ligand 1 (PD-L1) and programmed death-1 receptor (PD-1), in the treatment of advanced and recurrent cervical cancer. PD-L1 is an immune checkpoint inhibitor peptide which binds to its receptor PD-1. PD-1 and PD-L1 are expressed by many tumour cells, promoting immune evasion, and they are thus important immunotherapy targets [128]. A phase 2 trial of an HPV-16 peptide vaccine (ISA101) combined with nivolumab, a PD-1 antibody, have shown encouraging results, with increased immune-mediated tumour suppression [129]. Other similar combinations have also shown promising results [130,131].

Therapeutic vaccines have shown moderate effectiveness against CIN and HPV-associated cancers, but this has been less than anticipated, especially given the high effectiveness of the prophylactic vaccine [132]. One proposed mechanism for increasing effectiveness is by broadening coverage to encompass antigens other than E6 and E7. E1, for example, is increasingly implicated in carcinogenesis. To explore this, two therapeutic vaccines ChadOx1-HPV and MVA-HPV, have been developed by Vaccitech Ltd. using a viral vector. In addition to E6 and E7, these vaccines target E1, E2, E4 and E5 from five high-risk HPV subtypes [133]. Researchers hope that by targeting additional oncoproteins, including more HPV subtypes and the use of a viral vector, these vaccines, which are currently in a clinical trial (NCT04607850), will achieve a higher clinical efficacy than other developed therapeutic vaccines.

## 12. Conclusions

We are now 16 years following the licensure of the first prophylactic HPV vaccine, and the efficacy, effectiveness and safety of the vaccine is no longer in question. The main factor limiting the vaccine’s impact is a deficient population coverage [134]. The addition of HPV vaccination to infant vaccination programmes has been widely suggested as one way of addressing this, given the historic success of paediatric immunisation programmes [135]. This would of course require further studies to determine the appropriate dosing and safety of the vaccine in a paediatric population, but may indeed help increase vaccine coverage, particularly in areas with limited resources.

Another interesting concept called FASTER suggests combining HPV vaccination with HPV-based screening in women aged up to 50 [136]. Using this model, women who test negative for HPV DNA can expect to benefit from a high vaccine efficacy, as observed in the per-protocol HPV-naïve cohorts in the vaccine trials. Women who are HPV-positive will undergo further triage or screen-and-treat protocols [136]. While this approach will have a high acceptance rate among patients, infrastructural constraints may make it challenging to introduce [137].

Equity of access to the HPV vaccine, particularly in LLMICs, is a prerequisite for advancing sustainable HPV vaccination globally. Affordability, political buy-in and addressing cold-chain challenges are all ways to achieve this. The local development of biosimilar vaccines, such as Cecolin^®^, is already helping to drive vaccine costs down and increase availability. In the long term, the development of additional cheaper vaccines with less cold-chain dependence will increase HPV vaccine coverage and success. Furthermore, the gradual endorsement of single-dosing schedules will further reduce costs and improve full-dose coverage globally.

From a racial and ethnic perspective, there remain concerns regarding vaccine equity given ethnic and racial variations in HPV genotype prevalence, which may limit the effectiveness of the vaccine in certain nonwhite populations [138]. The complexity and cost of including more subtype VLPs would make addressing this concern challenging using the current vaccine instrument, and an alternative approach could be to identify a single antigenic target common to all HPV subtypes [134].

Cervical screening of vaccinated women is still recommended, as there remains the potential for disease caused by nonvaccine oncogenic HPV subtypes. Where it was previously theorised that vaccination against the common oncogenic HPV subtypes would result in an increase in the prevalence of the less common, nonvaccine oncogenic subtypes, early studies indicated that there was no evidence of “type replacement”, i.e., the replacement of the common oncogenic HPV subtypes with the less common ones [139]. Due to the reduced prevalence of CIN2+ in vaccinated women, however, cervical screening recommendations may need to be altered in the long term as vaccine coverage increases, to achieve maximal cost-effectiveness [134].

The comprehensive approach to cervical cancer control consists of a triad of primary, secondary and tertiary preventative measures. With the highly efficacious prophylactic vaccine, existing cervical screening measures and an anticipated effective therapeutic vaccine, alongside established treatment methods, achieving cervical cancer elimination is within reach. A focus on increasing the coverage and uptake of the HPV vaccine globally will certainly accelerate the attainment of this goal.

## Figures and Tables

**Figure 1 diagnostics-13-00243-f001:**
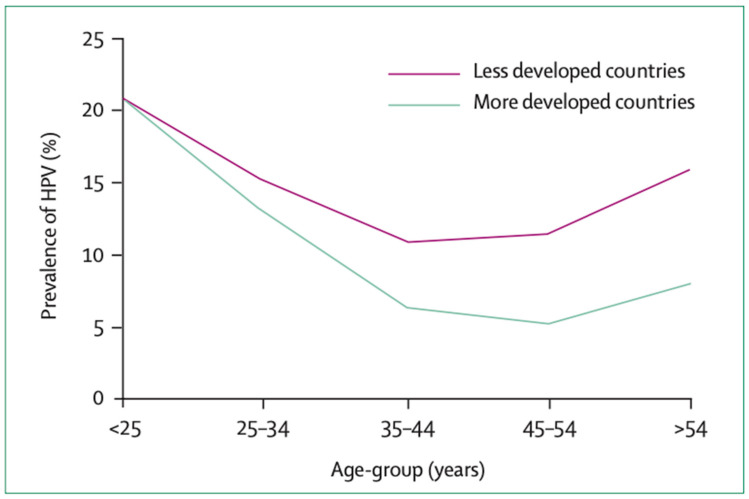
Prevalence of HPV worldwide by age (reproduced with permission from de Sanjosé et al. [11]).

**Figure 2 diagnostics-13-00243-f002:**
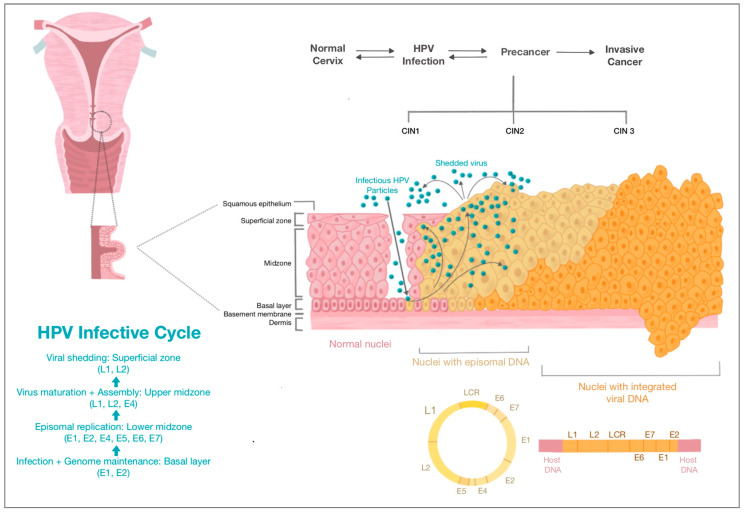
HPV infective cycle and cervical carcinogenesis (Source: authors).

**Figure 3 diagnostics-13-00243-f003:**
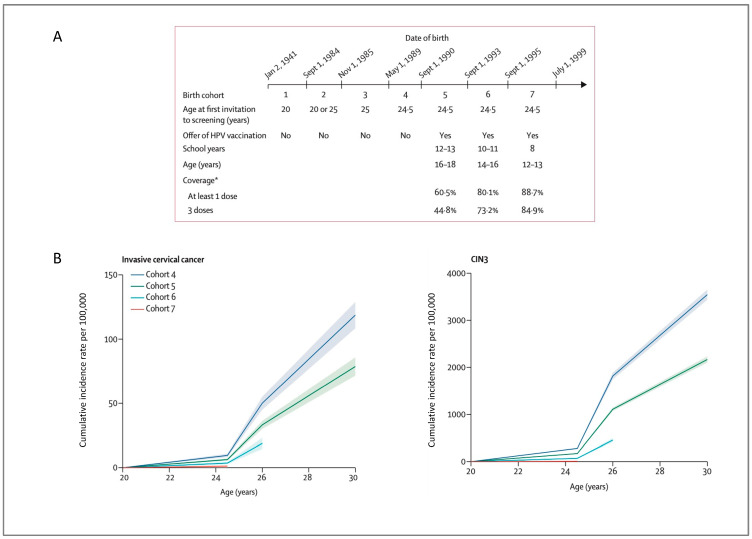
Cumulative incidence of cervical cancer and CIN3 using data from England showing a reduced incidence of cervical cancer and CIN3 in vaccinated cohorts (reproduced with permission from Falcaro et al. [76]) (**A**), schematic representation of birth cohorts [76] * Vaccine coverages include (where data are available) mop-up vaccinations (i.e. when females are vaccinated in a later year than the one in which they were first offered vaccination [76] (**B**), Cumulative incidence rates of invasive cervical cancer and CIN3 by birth cohort [76].

**Figure 4 diagnostics-13-00243-f004:**
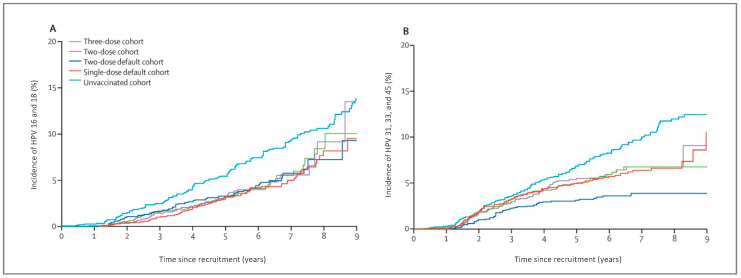
Incidence of HPV-16 and HPV-18 (**A**) and HPV-31, HPV-33 and HPV-45 (**B**), after one, two, three or no doses of the quadrivalent vaccine, showing similar HPV infection incidence in the one-, two- and three-dose cohorts (reproduced with permission from Basu et al. [84]).

**Figure 5 diagnostics-13-00243-f005:**
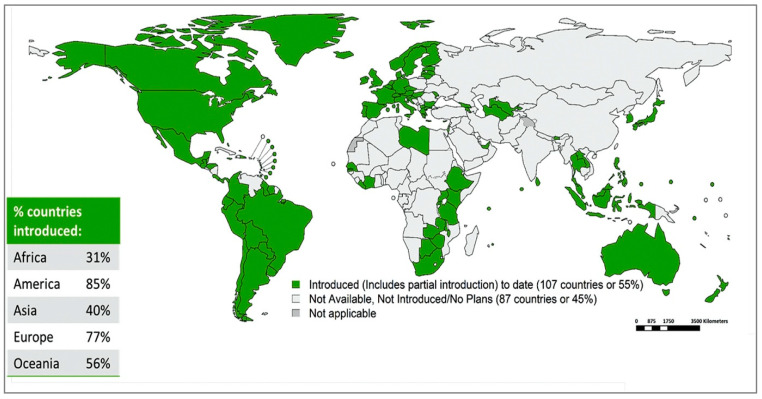
WHO member states with HPV vaccination in national immunisation schedule. (Reproduced with permission from Bruni et al. [111]).

**Table 1 diagnostics-13-00243-t001:** Characteristics of currently licensed prophylactic HPV vaccines.

Vaccine Brand Name	Valency and VLP Types	Manufacturer and Licensure Date	Adjuvant	Expression System
Gardasil^®^	Quadrivalent HPV-6, HPV-11, HPV-16, HPV-18	Merck & Co. 2006	Amorphous aluminium hydroxyphosphate sulphate 225 µg	Yeast *Saccharomyces cerevisiae* expressing L1
Cervarix^®^	Bivalent HPV-16, HPV-18	GlaxoSmithKline 2007	AS04 0.5 mg aluminium hydroxide and 50 µg 3-0-desacyl-4’ monophosphoryl lipid A	Insect cell line infected with recombinant baculovirus encoding L1
Gardasil 9^®^	Nonavalent HPV-6, HPV-11, HPV-16, HPV-18, HPV-31, HPV-33, HPV-45, HPV-52, HPV-58	Merck & Co. 2014	Amorphous aluminium hydroxyphosphate sulphate 500 µg	Yeast *Saccharomyces cerevisiae* expressing L1
Cecolin^®^	Bivalent HPV-16, HPV-18	Xiamen Innovax Biotechnology 2020	Aluminium hydroxide 208 µg	*Escherichia coli* expressing L1
Walvax recombinant HPV vaccine	Bivalent HPV-16, HPV-18	Shanghai Zerun Biotechnology (Subsidiary of Walvax Biotechnology) 2022	Aluminium phosphate	Yeast *Pichia Pastoris* expressing L1

## Data Availability

Not applicable.

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
