# Peer review of "Updates on HPV Vaccination"

_diagnostics, 2023, doi:10.3390/diagnostics13020243_

Round 1
Reviewer 1 Report
I believe the manuscript well summarizes existing evidence on HPV. The scoping review, although not conducted using systematic search, provides sufficient scientific summaries of various topics related to HPV. The manuscript will be helpful for those who are new to the field or a good starting place for those who are conducting reviews. While other scoping reviews on this topic exist, the manuscript updates the current evidence and has own value.
Author Response
Many thanks for your comments
Reviewer 2 Report
The authors present a comprehensive overview over HPV vaccines, their efficacy, effectiveness and safety, and implementations worldwide. Such an overview, covering all aspects of HPV vaccines in one paper, I think will be very much welcomed in the research community.
I have two smaller comments regarding the content.
1) The oncogenicity of the HPV types are briefly described in lines 65-66. I would have included one or more references about the issue. Also, HPV16 is by far the most oncogenic HPV type for cervical cancer, as well as the other HPV-related cancers, and should be mentioned.
2) There is an interesting concept called "FASTER" elimination of cervical cancer, suggesting concomitant screening and HPV vaccination, to maximize the combined impact of both measures, which could be mentioned.
There are some miss-spellings.
3) Vulval cancer (should be vulvar) (line 64)
4) In "early proteins W1, E2, E4, E5 and E6 and E7", I assume W1 should be E1 (line 79)
5) In "Two randomised control trials also showed reduced CIN recurrence rated in vaccinated women"
rated should be changed to rates (line 97).
The following sentence (lines 251-253) is hard to understand and should probably be rewritten:
6) "Although the nonavalent Gardasil 9® vaccine did not undergo prelicensure trials in men, immunogenicity studies showed a similar immune response against HPV-6, HPV-11, HPV-16 and HPV-18 was elicited in men when compared to the quadrivalent Gardasil® vaccine
Referencing to others work:
7) Figure 2: Have the authors made it themselves. If not, the source should be referred to.
Author Response
The authors present a comprehensive overview over HPV vaccines, their efficacy, effectiveness and safety, and implementations worldwide. Such an overview, covering all aspects of HPV vaccines in one paper, I think will be very much welcomed in the research community.
Response: Many thanks for your comments.
Point 1: The oncogenicity of the HPV types are briefly described in lines 65-66. I would have included one or more references about the issue. Also, HPV16 is by far the most oncogenic HPV type for cervical cancer, as well as the other HPV-related cancers, and should be mentioned.
Response 1: Paragraph changed to below, with additional references included:
There are over 100 subtypes of HPV, characterised into high-risk and low-risk subtypes depending on oncogenic potential. Low-risk HPV subtypes include HPV 6, 11, 42, 43 and 44 [12]. HPV 6 and 11 are the most prevalent non-oncogenic subtypes and are responsible for over 90% of all cases of genital warts [13]. The oncogenic subtypes include HPV 16, 18, 33, 35, 45 and 58, and these have the potential to cause cervical, oropharyngeal, vaginal, vulvar, penile and anal cancers [12], [14]. Cervical cancer is by far the most prevalent HPV-associated cancer, and HPV 16 is the most common causative subtype, followed by HPV 18 [14]. Jointly HPV 16 and 18 account for approximately 70% of all cervical cancer cases [9].
There may be racial and ethnic differences in age at peak HPV incidence, degree of HPV persistence and prevalence of high-risk HPV subtypes [14], [15], [16] and a number of studies have looked into these differences by region. Some recent studies have suggested a higher burden of non-HPV 16 or 18 in Black and African women with cervical cancer [17], [18]. While this requires further studies, HPV 16 and 18 remain the most common high-risk subtypes associated with cervical cancer worldwide [14], [19].
Point 2: There is an interesting concept called "FASTER" elimination of cervical cancer, suggesting concomitant screening and HPV vaccination, to maximize the combined impact of both measures, which could be mentioned.
Response 2: Many thanks for highlighting this concept. I have included a paragraph on this in the conclusions section of the article:
Another interesting concept called FASTER suggests combining HPV vaccination with HPV-based screening in women aged up to 50 [136]. Using this model, women who test negative for HPV DNA can expect to benefit from high vaccine efficacy, as observed in the per-protocol HPV-naïve cohorts in the vaccine trials. Women who are HPV positive will undergo further triage or screen-and-treat protocols [136]. While this approach will have a high acceptance rate among patients, infrastructural constraints may make it challenging to introduce [137].
Point 3: Misspelling 1: Vulval cancer (should be vulvar) (line 64)
Response 3: Changed to vulvar
Point 4: Misspelling 2: In "early proteins W1, E2, E4, E5 and E6 and E7", I assume W1 should be E1 (line 79)
Response 4: Changed to E1
Point 5: Misspelling 3: In "Two randomised control trials also showed reduced CIN recurrence rated in vaccinated women" rated should be changed to rates (line 408).
Response 5: Changed to rates
Point 6: The following sentence (lines 251-253) is hard to understand and should probably be rewritten: Although the nonavalent Gardasil 9® vaccine did not undergo prelicensure trials in men, immunogenicity studies showed a similar immune response against HPV-6, HPV-11, HPV-16 and HPV-18 was elicited in men when compared to the quadrivalent Gardasil® vaccine
Response 6: Changed to read:
Unlike the quadrivalent Gardasil® vaccine, the nonavalent Gardasil 9® vaccine did not undergo clinical trials in men prior to licensure. Rather, immunogenicity studies were done which showed that the nonavalent vaccine elicited immune responses similar to the quadrivalent vaccine against HPV-6, HPV-11, HPV-16 and HPV-18 [59]. Based on this and other safety data, Gardasil 9® has been licensed for use in men [60]. In the US, the FDA approved its indicated use in the prevention of HPV-associated oropharyngeal and head and neck cancers; a randomised controlled trial is underway to evaluate the efficacy of the Gardasil 9® vaccine against persistent oral HPV infection [61].
Point 7: Referencing to others work: Figure 2: Have the authors made it themselves. If not, the source should be referred to.
Response 7: Authors created this figure. Have included “Source: authors’